# WikiDBs: A Large-Scale Corpus of Relational Databases from Wikidata

**Liane Vogel,[1] Jan-Micha Bodensohn,[2,1] Carsten Binnig[1,2]**
[1]Technical University of Darmstadt, Germany
[2]German Research Center for Artificial Intelligence (DFKI), Darmstadt, Germany

## Abstract

Deep learning on tabular data, and particularly tabular representation learning, has recently gained growing interest. However, representation learning for relational databases with *multiple* tables is still an underexplored area, which may be attributed to the lack of openly available resources. To support the development of foundation models for tabular data and relational databases, we introduce **WikiDBs, a novel open-source corpus of 100,000 relational databases**. Each database consists of multiple tables connected by foreign keys. The corpus is based on Wikidata and aims to follow certain characteristics of real-world databases. In this paper, we describe the dataset and our method for creating it. By making our code publicly available, we enable others to create tailored versions of the dataset, for example, by creating databases in different languages. Finally, we conduct a set of initial experiments to showcase how WikiDBs can be used to train for data engineering tasks, such as missing value imputation and column type annotation.

## 1 Introduction

**The importance of representation learning on relational data.** Foundation models for text, images, and videos support users in many every-day tasks, reducing the manual effort required for end tasks such as machine translation, image generation, and automated software development [13].

While research on learning such foundation models is currently dominated by text and images, considerable progress has recently been made on other modalities such as tabular data [14, 6]. This is important since a non-negligible amount of data is expressed in tabular form, in particular enterprise data [11]. Pre-trained foundation models that are specialized to handle relational data could reduce the manual effort required for many data-heavy tasks such as data integration and data cleaning [47].

Over the last years, several deep learning-based approaches have been developed to solve data engineering tasks like entity matching [38] and missing value imputation [35] on individual tables. These efforts have been supported by various large-scale table corpora, such as GitTables [23] and WikiTables [8], which provide the training data required to train these deep learning models.

**Lack of large corpora for relational databases.** Considerable amounts of tabular data, especially in enterprise contexts, are not stored in individual self-contained tables, but rather in relational databases consisting of multiple tables that are closely connected by foreign key relationships. For these types of relational databases, however, there is a lack of large-scale openly available training data as well as deep neural network architectures that can process multiple related tables. Collecting large corpora of such relational data is not trivial. Due to the sensitivity of the data stored in relational databases, real-world enterprise databases are typically highly confidential and therefore not accessible to the scientific community, resulting in a lack of openly available corpora.

38th Conference on Neural Information Processing Systems (NeurIPS 2024) Track on Datasets and Benchmarks.

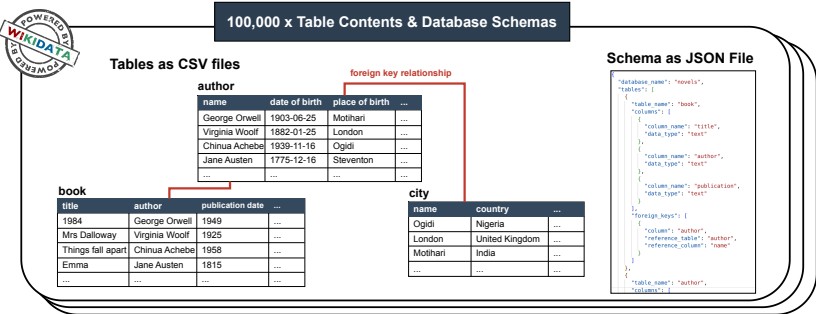

Figure 1: We release **WikiDBs**, a dataset of 100,000 relational databases based on data from Wikidata. The figure shows one of the database schemas with examples of the table content (left) and how the schema is stored as a JSON file (right).

**The need for real-world data.**    As a consequence, in the field of database research, it is common to use synthetic databases such as the datasets in the TPC benchmarks [26, 40]. While this approach may be sufficient for testing database internals, representation learning on relational databases requires a large variety of databases with diverse content from multiple domains, which the existing benchmarks are too few and not diverse enough to provide. In addition, these usually synthetically generated datasets often contain only numerical and categorical data, but not meaningful text and context as contained in real-world databases. However, according to [50], a significant fraction of the data in databases is stored as text. In order to have realistic training data, it is thus important to obtain databases that contain not only numerical and categorical, but also textual data.

**Towards a new corpus of relational databases.**    We aim to foster research on foundational models for relational data by creating a new, large-scale resource. Our goal hereby is to have realistic data that is grounded in real-world knowledge instead of synthetically generated data. While there are some publicly available relational databases such as the Internet Movie Database (IMDb) and the MIMIC database [29], there is so far no large corpus containing many relational databases. Therefore, we present an approach that uses the Wikidata knowledge base [2], which is openly available with a CC0 license, as the basis for deriving a large corpus of relational databases.

Along with this paper, we release a large-scale, open-access dataset called *WikiDBs* – a corpus of 100,000 relational databases extracted from Wikidata – covering a broad spectrum of diverse domains. WikiDBs extends the methodology presented in WikiDBs-10k [48]. Compared to WikiDBs-10k, the WikiDBs corpus presented in this work includes databases with up to 100 tables as well as tables of larger sizes. Furthermore, the created databases are more coherent in terms of the domains they cover, and the table and column names are more diverse.

We make our dataset and code publicly available under the CC BY 4.0 license.[1] Moreover, we encourage others to create customized versions of the WikiDBs corpus, for example, by focusing on tables in other languages apart from English or with different data characteristics such as table sizes or data sparsity. We are releasing the dataset of 100k databases on Zenodo.[2] The data is provided in five compressed splits of 20k databases each. Each split is 10GB in size (33GB uncompressed); the complete uncompressed dataset requires approximately 162 GB of disk space.

**Contributions.**    In this paper, we describe our methodology for creating databases that are grounded in knowledge from Wikidata in Section 3. In Section 4, we describe the WikiDBs corpus in detail and compare its characteristics to statistics available for real-world relational databases. Finally, in Section 5, we present a set of initial experiments to showcase how our corpus can be used to learn representations that are informed by multiple tables in a relational database.

To summarize, this paper makes the following contributions: (1) We introduce a novel method of extracting multi-table relational databases from Wikidata, which we make publicly available for others to generate customized relational database corpora. (2) We release a large-scale corpus of 100,000 relational databases, which we compare to existing tabular datasets as well as statistics about

---

real-world relational databases. (3) We showcase how our dataset could be used to train foundation models for relational data by conducting a first set of experiments on the data engineering tasks of missing value imputation and table and column name inference. Additionally we show how pre-training on WikiDBs benefits a downstream task.

We see our work as a first and important step to enable Large Database Models (LDMs) – foundation models trained on large corpora of structured data rather than text or images – that can be used in a task-agnostic manner to tackle a broad set of tasks on relational data including data cleaning, data integration, data transformation, question answering, table retrieval, and many more.

## 2 Related Work

**Creating tables from Wikidata.** Closest to our work is the WikidataTables2023R1 dataset [21] created for the SemTab challenge [28], which consists of around 10,000 tables created from Wikidata. In the SemTab challenge, this tabular data is matched to knowledge graphs. Compared to our work, the tables in the WikidataTables2023R1 dataset are considerably smaller in terms of rows and columns (on average only 2.5 columns and 4.7 rows per table), and the dataset contains only individual tables instead of relational databases.

**Single-table repositories.** So far, tabular representation learning focuses primarily on learning representations of individual self-contained tables. Commonly used corpora include GitTables [23], WikiTables [8], the Dresden Web Table Corpus (DWTC) [17], and the WDC corpora [33]. Currently, the largest collection of tables is the TabLib dataset [18], which contains 627 million tables extracted from GitHub and CommonCrawl.

**Multi-table repositories.** There are only few datasets containing relational databases with multiple connected tables. To support machine learning on multi-table relational data, [37] publish the CTU Prague Relational Learning Repository, which currently includes 83 databases. The SQLShare corpus [27] is a query workload dataset which includes 64 databases collected from real-world users (primarily researchers and scientists) from the SQLShare webservice [1]. Both repositories are thus much smaller than the corpora of individual tables that are commonly used for table representation learning. Finally, the GitSchemas [16] repository contains 50k database schemas based on SQL files from public GitHub repositories, thus providing highly relevant insights into real-world databases. However, the corpus does not include the actual content of the databases, making it unsuitable to train multi-table foundation models. As a concurrent work to ours, [15] extend GitSchemas and release the SchemaPile Dataset, which contains around 220,000 database schemas from GitHub, some of which also include the table contents.

**End task datasets based on Wikidata.** Apart from the SemTab challenge [28] mentioned before, Wikidata has been used to build datasets for named entity classification [20] and named entity disambiguation [12] as well as complex sequential question answering [43]. Moreover, [5] verbalize knowledge graph triples from Wikidata, and [19] create alignments between Wikidata triples and Wikipedia abstracts.

## 3 Construction Methodology

Figure 2 gives an overview of the necessary steps to create a database based on the data in Wikidata. First, as a one-time effort, we load and pre-process the Wikidata JSON dump. Next, we build the individual databases in a table-by-table manner. In order to build databases that are coherent with respect to a particular topic domain, like *biology*, *movies*, or *sports*, we make use of semantic representations. Finally, since the properties from Wikidata are not expressive enough to serve as table and column names, we use GPT-4o to paraphrase them to better match the topic domain of each database.

For each database, we store the individual tables in the CSV format and additionally provide the schema information, which includes the table structure and the foreign keys (shown in Figure 1, right), as well as the original table and column names from Wikidata as alternative names.

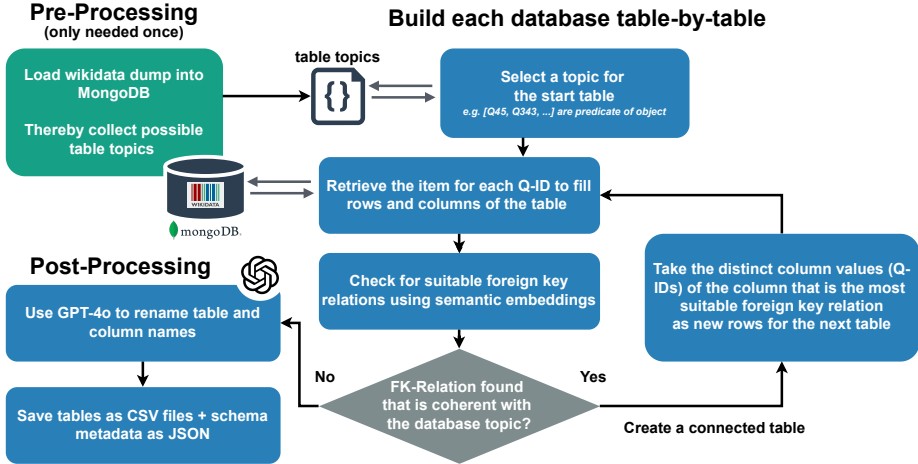

Figure 2: Construction methodology of WikiDBs.

## 3.1 Wikidata Format and Pre-Processing

The data in Wikidata is stored in a document-oriented database, where documents represent items that are instances of different concepts, such as *artists* or *paintings*. In this way, concepts closely resemble the notion of tables.

Every item in Wikidata is associated with a unique identifier, the so-called *QID*. The item representing the NeurIPS proceedings for example has the id *Q15767446*. Properties of items are stored in the form of key-value pairs, where property names are saved with their corresponding value. Most important are properties (e.g. the *main subject (P921)*) that resemble attributes of a table row. Moreover, properties also store other information such as links to related concepts of an item. For example, the NeurIPS proceedings item has the property *instance of (P31) scientific journal (Q5633421)*.

We analyze all Wikidata items regarding such properties that link to other Wikidata items to collect potential table topics. During the pre-processing phase, we build up a lookup structure which maps concepts (i.e. tables) to potential items (i.e., rows). The values of each item correspond to the rows of the tables, the properties of the values form the column headers. The lookup structure allows us to select start tables for databases in a targeted manner, especially ensuring that they have a minimum number of rows.

We load the Wikidata JSON dump [4] into a MongoDB database, which allows for efficient querying of the data. We also experimented with using the Wikidata SPARQL API. However, the query runtime limits prevent the creation of a large dataset.

## 3.2 Building Coherent Databases

As mentioned above, we create each database in a table-by-table manner, beginning with a start table that we later extend by multiple connected tables.

**Creating individual tables.** The creation of a relational table from Wikidata is thus made possible by the properties of items in Wikidata. The information that the NeurIPS proceedings item is an instance of *scientific journal* allows us to search Wikidata for all other items that are also tagged with the information that they are an instance of *scientific journal*.

A challenge in Wikidata is that every item (e.g., each journal) might use a different set of properties. For example, for some journals the main subject of the journal is available, while for others it is not. For constructing tables, we use the union of all properties. If a value is missing, we store a NULL-value in the table row. To avoid constructing tables with highly sparse columns, we prune columns of a table where the fraction of NULL-values is beyond a configurable threshold.[3]

**Creating coherent databases.** For each created table, some columns contain references to concepts that are also saved as items in Wikidata (see Figure 3). We use those columns that contain Wikidata

---

[3]For the corpus released with this paper, we prune columns with more than 50% NULL-values.

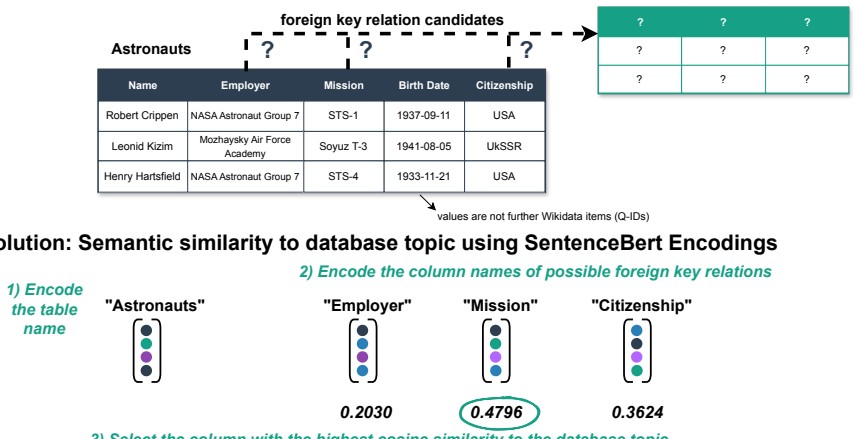

**Problem: Which column is most suitable to create the next table of the database?**

**Solution: Semantic similarity to database topic using SentenceBert Encodings**

Figure 3: Building coherent databases by selecting semantically similar relations.

items to build further tables for the database. When constructing relational databases, we observe that selecting a concept in Wikidata as a starting point and then traversing relationships to other tables randomly will result in many topic shifts within each database since the connected tables may not match the topic of the original start table. To overcome this problem, we make use of the semantic similarity between the column names and a database topic embedding to select the most suitable columns for outgoing relations.

In the example shown in Figure 3, the table of *astronauts* contains the columns *employer*, *mission*, and *citizenship* that all contain references to other items, and are thus candidates to build a connected table with. In order to decide which of these relations is most suitable to stay coherent with the database topic, we represent each candidate column name using Sentence-BERT [42] and compute the cosine similarity to the database embedding. The database embedding is initialized with the embedding of the table name of the start table and updated with the embedding of every connected table added to the database. We always build the next table from the relation that has the highest cosine similarity to the database embedding. Furthermore, if all similarity scores are below a given threshold (we use *0.2*), we declare the database as completed. In our example, this method allows us to add the table of *missions* (linked via a foreign key to the table *astronauts*), where each row corresponds to a space mission and the columns contain information such as the launch date and the name of the used spacecraft.

### 3.3 Improving Table and Column Names

The column names of the tables in WikiDBs are the property names of the properties of each row item (e.g., *employer* and *date of birth*). In total, Wikidata contains only $11,887$ different properties. Given that our dataset includes thousands of databases with millions of columns, the property names used as column names are bound to repeat themselves. Moreover, the property names in Wikidata are often rather generic and thus not expressive in the context of each database. Likewise, the table names derived from Wikidata objects and properties are either (1) too specific in the case of the start table (e.g., *based on Winter landscape with skaters and bird trap* instead of *painting*), or (2) too generic in the case of the connected tables (e.g., *person* instead of *employee*). Therefore, to better reflect the characteristics of real-world databases, as part of post-processing (see Figure 2), we use a Large Language Model (GPT-4o[4] in our case [39]) to paraphrase table and column names to increase the variety of names.[5]

We prompt the language model once per table in the database and always include the table header with two randomly sampled rows serialized in CSV format. As shown in Figure 4, we first instruct

---

[4]We use the `gpt-4o-2024-08-06` version of the model with the temperature set to 0.

[5]We provide both paraphrased and original names in the corpus, allowing users to make a choice based on their use case.

```
Imagine a database with the topic: {{database_start_table_name}}
Please complete the following list of tasks, answering accurately and responding
↪   only with the required term or format.
1. Give an appropriate name for the database
Here is one of the tables in the database with some sample rows:
{{start_table}}
2. Specify an appropriate and realistic name for the given table.
3. Since the column names are rather generic, please find a name for each column
↪   that is more realistic and related to the database topic domain
↪   {{database_start_table_name}}.
If there is no better name, keep the original attribute name.
Respond with a JSON object that contains the improved database name, improved table
↪   name, and list of improved column names.
```

Figure 4: Prompt used to paraphrase the start table of each database.

the model to specify a name for the database and the start table, as well as to paraphrase all columns of the start table. We use OpenAI's Structured Outputs to specify the JSON schema of the desired output. After paraphrasing the start table of the database with the first prompt, we paraphrase the table and column names of all connected tables using the prompt shown in Figure 8 in the Appendix. In case the language model returns duplicate column names for a particular table,we keep the original names of the corresponding columns. Likewise, we resort to the original table names in case there are duplicates for a particular database.

Overall, the paraphrasing allows us to significantly increase the diversity of the table and column names, as we show in Section 4.

### 3.4 Parameters and Customizability

As mentioned, we make our code publicly available on GitHub under the CC-BY license. Our method includes a variety of parameters that enable the customization of the created databases, for example regarding the minimum and maximum number of tables per database, number of rows per table, and the sparsity of the columns. Additionally, our method for selecting the most coherent connected tables using Sentence-BERT [42] can be customized, for example by tweaking the similarity threshold for including connected tables. Furthermore, while we focus on English-language content in this corpus, our method can easily be used to create databases in other languages included in Wikidata.

In the following section, we describe the corpus released with this paper in more detail.

## 4 Dataset Characteristics

For our WikiDBs dataset, we want the characteristics of the data to reflect the properties of real-world databases. As enterprises do not share the statistics of their databases, we have to rely on the characteristics of available public resources and model our dataset in a similar way. In Table 1, we have collected characteristics of existing public resources, such as the number of tables in a database, and the average number of columns and rows per table.

For deriving statistics of other relational databases, we found only two existing collections, the Relational Learning Repository from CTU Prague [37] (which also includes TPC-H and IMDb) and the SQLShare [27] repository. However, all these repositories include only a small number of relational databases. Therefore, we also include the statistics of the significantly larger datasets GitSchemas [16] – which only contains schema information – and GitTables [23] – a corpus of individual tables. WikiDBs includes on average a higher number of columns per table than the other corpora, which is in line with findings about real-world enterprise databases, which also often include a large number of columns [27].

### 4.1 Dataset Statistics

**Data dimensions.** As shown in Table 1, our dataset consists of 100,000 databases that each have between two and 100 tables connected via foreign keys. In total, our dataset contains around 1.6M tables. On average, each table has 52.72 columns and 118.16 rows. Figure 5 shows the distributions

Table 1: Characteristics of existing resources compared to our new dataset. We report the median number of tables per database, as well as the average number of columns and rows. GitSchemas [16] does not contain the content (=rows) of the databases and GitTables [23] consists of single tables, not databases.

| | includes schema | includes table content | #DBs | #Tables | #Tables per DB | #Colums | #Rows |
|---|---|---|---|---|---|---|---|
| | | | | | Median | Avg. | Avg. |
| CTU Prague [37] | ✓ | ✓ | 83 | 813 | 5 | 6.0 | 4.8k |
| SQLShare [27] | ✓ | ✓ | 64 | 3.9k | 4 | 18.7 | 11k |
| GitSchemas [16] | ✓ | ✗ | 156k | 1.2M | 4 | 5.7 | - |
| GitTables [23] | ✗ | ✓ | - | 1M | 1 | 12.0 | 142 |
| WikiDBs-10k (ours) | ✓ | ✓ | 10k | 42.5k | 4 | 17.9 | 46 |
| **WikiDBs (ours)** | ✓ | ✓ | **100k** | **1.6M** | **3** | **52.7** | **118** |

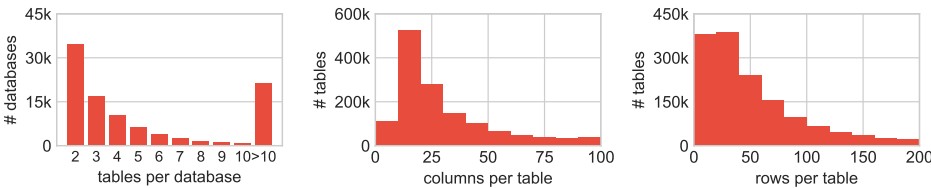

Figure 5: Data dimensions: tables per database, rows per table, and columns per table.

of the numbers of tables per database, rows per table, and columns per table. We observe that our corpus includes a broad variety of database and table sizes, which is in line with real-world relational databases. For example, 26% of the schemas in the GitSchemas dataset [16] include only two tables, whereas enterprise databases often include hundreds of tables.

**Data characteristics.** Figure 6 shows additional characteristics of the data in WikiDBs. First, we see that the table sparsity defined as the fraction of empty cells ranges from 0 to 0.5, a characteristic which can be tailored by tweaking our dataset generation parameters. Likewise, we observe that a large fraction (75.41%) of the columns store non-numerical values. Finally, paraphrasing the table and column names using a Large Language Model substantially increases the number of unique table and column names across all databases in the corpus, allowing us to significantly increase the diversity of such schema information.

**Links to Wikidata.** In addition to the cell values, our corpus includes the identifiers of the Wikidata entries for all cells corresponding to Wikidata items. This opens up the opportunity to adapt our corpus for a variety of table-based end tasks such as schema matching, entity matching, and deduplication.

## 4.2 Limitations of WikiDBs

As our corpus is grounded in Wikidata, the underlying data may potentially be noisy, untruthful, and hard to attribute to individual authors. Moreover, we observe that the coverage of world knowledge in Wikidata is highly skewed towards communities (e.g., astronomy) that are more inclined to enter their knowledge into structured knowledge bases. Since our automatic paraphrasing procedure is based on Large Language Models, it is vulnerable to their well-known weaknesses, including hallucinations and social biases [7].

Furthermore, table sizes in WikiDBs are still rather small, as on average each table has only 118 rows. During construction we already aim to maximize table sizes, but Wikidata does not provide more data to construct larger tables. One interesting observation, however, is that Wikidata is far from complete and many entries of less well-known entities are missing. As such, a potential avenue is to add more data by trying out data augmentation strategies on the databases in WikiDBs.

Finally, the characteristics of real-world databases can be quite diverse [10], especially in terms of data types, cell values, or column names, and not all of them are reflected in WikiDBs. Creating subsets by e.g. filtering for only numerical columns or increasing the sparsity is an interesting future avenue.

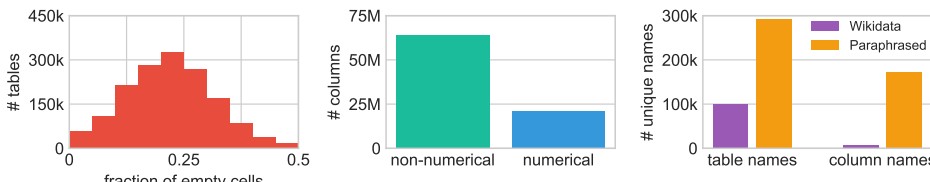

Figure 6: Data characteristics: table sparsity, data types, and diversity of table and column names.

# 5 Initial Experiments

We hope that our corpus fosters more research on models for table representation learning that can take data from multiple connected tables into account. In the following, we empirically demonstrate the usefulness of our corpus with initial experiments on WikiDBs.

We present results for three different data-engineering tasks, namely predicting missing values, column names, and table names. We chose these tasks because they require an understanding of table structures at different granularities: While cell value imputation requires context from data in the same row and column, table name prediction can benefit from the entire table as well as from connected tables in a database. In addition, we investigate how pre-training on WikiDBs benefits downstream tasks using the task of semantic column type annotation as an example.

Predicting missing values, column names and table names are well-established tasks in the area of data engineering. For example, predicting column names is important when schema information must be derived from data (e.g., a CSV file without a header). This task is also closely related to other established tasks such as annotating semantic column types for data search and discovery [45, 24]. We model all these tasks as generative tasks where the missing information is generated by the model token per token, rather than as classification tasks (e.g. where possible column names are known upfront) in order to be able to work with unseen data.

## 5.1 Pre-training Procedure

It is not well understood yet what model architectures and sizes are required for tabular foundation models. We choose task formulations and model architectures based on related work in tabular representation learning [46, 52]. We use BART [34] and T5 [41] because they are encoder-decoder language models, and as such, enable us to solve generation-based downstream tasks, such as generating unseen missing values. Additionally, they are available in different sizes. We therefore think that they are a good starting point for tabular foundation models.

We split the databases from a preliminary version of our dataset into 70/10/20 percent for training, validation and testing. We fine-tune pre-trained T5 models of different sizes ([41]) from the Huggingface library [51] on tables from our dataset using LoRA [22] with $r$=64 and $alpha$=128. In line with related work such as TaBERT and RPT, the model receives tables with linearized table rows as an input and is trained to generate the missing table name, column name, or cell value as an output. We train for 50 epochs with an initial learning rate of $3.2e - 4$ and a cosine annealing schedule to reconstruct masked table names, column names and cell values. The checkpoint with the best accuracy on the validation set is used for evaluation. We train our models using an effective batch size of 128, employing gradient accumulation based on available compute hardware (between 8 and 64 A100 GPUs).

## 5.2 Experiment 1: Fine-tuning Language Models on Tabular Data

As a first experiment, we compare the performance of T5 models of different scales on our new dataset before and after our table and column name rephrasing step (see Table 2). The results show that the names extracted from Wikidata properties are considerably less diverse and therefore much easier to predict. As seen in Figure 6 (right), rephrasing vastly increases the amount of unique column names in the dataset, making them harder to predict. Additionally, we observe that scaling the model increases the performance on all tasks, however there are diminishing returns and the computational costs of the models vastly outpace the increase in performance. This underlies our motivation for providing a resource to foster the development of more suitable model architectures that can efficiently utilize the relational structure of databases.

Table 2: Results of T5 models of different scales on our WikiDBs dataset before and after paraphrasing table and column names. We report token-wise accuracy and exact (string) match scores.

| Model | Train Data | # Params | Accuracy [%] | | | Exact Match [%] | | |
|---|---|---|---|---|---|---|---|---|
| | | | Cell | Column | Table Name | Cell | Column | Table Name |
| T5-base$_{table}$ | WikiDBs | 220M | 46.87 | 69.44 | 39.03 | 38.40 | 67.20 | 26.40 |
| T5-large$_{table}$ | (table/column names | 770M | 49.00 | 75.22 | 41.05 | 40.20 | 73.80 | 29.40 |
| T5-3B$_{table}$ | from wikidata) | **2.8B** | **54.19** | **79.41** | **42.68** | **45.00** | **78.40** | **29.60** |
| T5-base$_{table}$ | WikiDBs | 220M | 39.80 | 47.53 | 23.69 | 31.40 | 40.40 | 15.40 |
| T5-large$_{table}$ | (GPT-4o paraphrased | 770M | 46.21 | 52.12 | 27.11 | 38.20 | 45.80 | 18.20 |
| T5-3B$_{table}$ | table/column names) | **2.8B** | **51.37** | **60.60** | **31.61** | **41.20** | **53.60** | **20.40** |

Table 3: Results of our T5 models on the WikiDBs10k [48] test set to provide a reference point for the capabilities of our T5-based models. Furthermore, a comparison to Table 2 highlights the increased difficulty of our new larger corpus.

| Model | # Params | Accuracy [%] | | | Exact Match [%] | | |
|---|---|---|---|---|---|---|---|
| | | Cell | Column | Table Name | Cell | Column | Table Name |
| BART$_{table}$ [46, 48] | 400M | 24.14 | 48.98 | 48.80 | - | - | - |
| BART$_{table}$+GNN$_{db}$ [48] | 400M | 30.22 | 69.37 | **50.08** | - | - | - |
| T5-base$_{table}$ (Ours) | 220M | 53.19 | 86.77 | 34.92 | 43.65 | 86.51 | 29.37 |
| T5-large$_{table}$ (Ours) | 770M | 58.62 | 90.87 | 42.89 | 50.00 | 89.68 | **38.10** |
| T5-3B$_{table}$ (Ours) | **2.8B** | **60.71** | **92.46** | 32.56 | **52.38** | **91.27** | 30.16 |

## 5.3 Experiment 2: Language Models Struggle to Use Relational Context

In Table 3 we compare different sizes of LM-based models to models specifically adapted to handle relational data. We apply the architecture introduced in [49], which is a combination of language models (LMs) and graph neural networks (GNNs). To provide more context and explicitly represent the structure of relational databases, tables and databases are modeled as graphs. In the LM+GNN model, the LM encoder is used to create initial encodings for each graph node, which are updated with information from neighboring nodes through GNN message passing (i.e., to propagate information along the table structure). Finally, we use the LM decoder to generate the masked value.

We see that a small language model (BART only) is considerably worse than combining the language model with a GNN (BART + GNN). Moreover, we see that larger language models clearly help since larger T5 models are considerably better than the smaller BART-based models. Overall, this calls for architectural improvements, such as combining larger language models with structured models such as GNNs, as an alternative to the exponentially costly path of naive scaling, which is an avenue which our corpus enables.

To further investigate the benefits of the multi-table contexts in WikiDBs, we train a single-table and a multi-table model using the GNN-based architecture from [49] with T5 as LM on the task of table name prediction. We initialize both models (single-table and multi-table) with the same pre-trained checkpoint of T5-3B and use a training setup similar to that described in Section 5.1 to train the GNN. For the training, we use a batch size of 32 and a learning rate of 3.2e-4 and train for 100 epochs.
In this preliminary experiment, our multi-table model already provides benefits over the single-table model without much tuning w.r.t. how many of the neighboring tables we add as context. To be more precise, the multi-table model achieves an exact match accuracy[6] of 18.40% and a token-wise accuracy of 28.34%, whereas the single-table model only achieves an exact match accuracy of 18.00% and a token-wise accuracy of 26.07%, already indicating the benefit of the multi-table context over the single-table context. We expect this gap to further increase as our understanding of multi-table model architectures and training processes improves.

---

[6]Exact match accuracy (EM accuracy) measures how often the predicted table name is exactly the same as the ground truth table name. By contrast, the more relaxed token-wise accuracy considers how many of the individual tokens of the predicted table name match those of the ground truth table name. For example, the predicted table name "BasketballPlayer" for the ground truth "BasketballPlayerStatistics" results in an EM accuracy of 0, but a token-wise accuracy of 0.67 since two of the three tokens are correct.

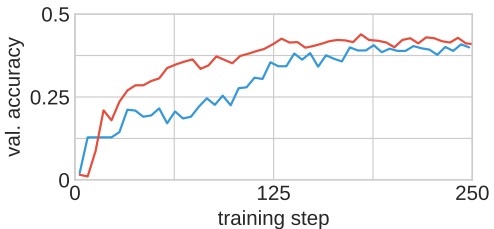

Figure 7: Column type annotation (CTA) validation accuracies of vanilla T5 and T5 pre-trained on WikiDBs during fine-tuning on the GitTablesCTA benchmark. The T5 model that has been pre-trained on WikiDBs requires fewer fine-tuning steps to achieve higher accuracies.

## 5.4  Experiment 3: Benefits of Pre-Training on WikiDBs for Downstream Tasks

In this experiment, we evaluate how pre-training on WikiDBs can be beneficial for downstream task applications. We compare T5 pre-trained on WikiDBs to vanilla T5 on the task of column type annotation (CTA), a task often used in the literature [24, 53, 32, 31] where the goal is to annotate each table column with a semantic column type from a pre-defined ontology (e.g., DBpedia [36]). Since there is no multi-table evaluation dataset for CTA, we use a corpus based on individual tables to show the effects of pre-training with WikiDBs.

In particular, for the experiment, we use the GitTablesCTA benchmark [25] also used in the 2021 SemTab challenge [3]. We pre-train T5-3B on WikiDBs as described in Section 5.1. Afterward, we fine-tune the pre-trained T5-3B as well as the vanilla T5-3B model for 50 epochs on GitTablesCTA to predict the column type of a specified column based on the first three rows of the table.

T5 pre-trained on WikiDBs achieves a validation accuracy of 44%, whereas vanilla T5 only achieves an accuracy of 41%. Figure 7 shows that T5, when pre-trained on WikiDBs, requires fewer fine-tuning steps to achieve higher accuracies. For example, after the first 100 training steps, T5 pre-trained on WikiDBs achieves a maximum validation accuracy of 37%, whereas vanilla T5 only achieves a validation accuracy of 25%.

## 6  Conclusion and Broader Impact

With the corpus presented in this work, we establish the foundations for research on Large Database Models (LDMs) – foundation models trained on large corpora of structured data rather than text or images. While there is already some initial work that has looked into model architectures for foundational models for tabular data (or sometimes also referred to as table representation learning), the corpora used in these works are significantly too small to enable LDMs. Moreover, as mentioned before, the existing corpora often cover only individual self-contained tables and thus do not allow LDMs to learn from tables and their context (i.e., neighboring tables), which often provide important signals for understanding the data in a table.

Furthermore, WikiDBs can be a useful resource for data discovery in data lakes [44] like table union and joinability search [30, 54] or table retrieval [9]. As WikiDBs includes the Wikidata IDs for every column and cell value, it can be integrated with existing resources in the semantic web community.

To showcase the value of the corpus, we have trained a first LDM using a hybrid model that combines an LM with a GNN to learn the inherent structure of relational data. The initial results showcase that when pre-training our model based on WikiDBs, we can efficiently solve various downstream tasks with the pre-trained model. However, we also think that we are still far away from an LDM for relational data that has the capacities that LLMs have today for text. As such, with the corpus in this paper we hope that we can foster more work in this direction and explore also different model variants for LDMs. Moreover, we also think that the corpus can be clearly further extended in the future to even larger and more diverse data sets. One interesting route is to use existing LLMs to generate relational databases from all data LLMs have seen during training.

## Acknowledgments and Disclosure of Funding

This work has been supported by the BMBF and the state of Hesse as part of the NHR Program and the HMWK cluster project 3AI. It was also partially funded by the LOEWE Spitzenprofessur of the state of Hesse. We also thank DFKI Darmstadt and hessian.AI for their support.

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

# 7 Appendix

```
Imagine a database with the topic: '{{database_start_table_name}}' and the
↪  database name '{{database_name}}'.
Please complete the following list of tasks, answering accurately and
↪  responding only with the required term or format.
Here is one of the tables in the database with some sample rows:
{{fk_table}}
1. The current name of the table is '{{fk_table_name}}'' and the table is
↪  referenced to from the following other tables: {{fk_relationships}}.
↪  Specify a better name for the given table, especially considering the
↪  content of the given table and the references {{fk_columns}}, or keep the
↪  current table name if it is already suitable.
2. Since the column names are rather generic, please find a name for each
↪  column that is more realistic and related to the table and database topic
↪  domain. If there is no better name, keep the original attribute name.
Respond with a JSON object that contains the improved table name and the list
↪  of improved column names.
```

Figure 8: Prompt used to rename every further table (apart from the start table) of each database.

