# OpenReview forum: "WikiDBs: A Large-Scale Corpus Of Relational Databases From Wikidata"
_NeurIPS.cc/2024/Datasets_and_Benchmarks_Track — NeurIPS 2024 Track Datasets and Benchmarks Spotlight_

### Official Review · Reviewer_nHYv · 2024-06-27
**Review of submission 1170**

**Rating:** 8
**Confidence:** 4

**Review:**

The paper is clearly written and concepts are explained well. The dataset introduced with the contribution will work as a significant baseline for increasing the effectiveness of LLMs when applied to tabular data, and in general for any problem that requires evaluating over databases that include multiple tables.

The main weaknesses of the paper lie in the small size of the tables (only few hundreds of rows on average), the experimental section (the scope of the experiments is quite small), and in some unconvincing choices made in the preparation of the database.

**Strengths:**

- The resulting dataset is large and represents a major improvement over previous baselines in the number of tables and databases.
- The preparation of the dataset is detailed and explained well.
- Experimental results showcase some interesting preliminary results about how the effectiveness of LLMs as they are applied to tabular data, and highlight that further work is needed to replicate the performance that is currently achieved on text.
- The dataset by itself will be a significant contribution not only to develop LLMs, but also for any studies more focused on RDBs.

**Additional Feedback:**

Please consider using parquet as a storage format, rather than CSV. Parquet files are compressed and store the schema used by the tables, whereas CSV files occupy far more space and do not retain information on the table schema.

The archive provided for the reviews is already quite sizeable, and the 100k version will be even harder to manage for a lot of users. I would suggest to keep the 20k version along with the larger 100k, and potentially release a smaller sample to help practitioners with "getting acquainted" with a smaller version of the dataset before working with the larger variants. I believe this would increase the reach of this contribution by covering different use cases.

**Clarity:**

The paper is well written and easy to read. Examples are clear and easy to understand.

**Correctness:**

Dataset preparation is well detailed and explained at a high level. Additional material addresses some points, however in general there is not enough information to properly reproduce the paper. Experiments appear correct.

**Documentation:**

The documentation provided in the repository is lacking. It is also unclear where the dataset is stored at all. No information on the resources required to download (disk space) or run the pre-processing (disk, CPU, RAM etc.) is provided, other than the run time on an unknown system. A license is provided. No code is available to reproduce the experiments in the repo version available at the time of this review.

**Ethics:**

No ethical concerns.

**Limitations:**

- While the dataset includes a very large number of databases and associated tables, the average size of the tables themselves is quite small, which is not very representative of tables in read RDBs.
- The method used to find candidates for each RDB is not convincing: cosine similarity does not guarantee that joins can be done, and it is further undermined by the fact that column names are not very informative. A better justification of the effectiveness (through experiments) or the evaluation of alternative methods would make the case for this choice stronger.
- The experiments do not appear to have large practical application. While I understand that the main argument of the paper is preparing the foundations for an LDM, I believe that evaluating the dataset from other points of view (e.g., information retrieval) may strengthen the submission.
- There is very little information on the physical resources required to use the dataset, and on those required to prepare it in the first place. The experimental section is similarly light on the details on the computing resources.

**Opportunities For Improvement:**

o1) While the overall size of the dataset is large, tables are quite small as the average number of rows is in the range of few hundreds. Typical relational databases tend to involve a much larger number of records, which is not reflected in the current version of WikiDBs. I imagine this is due to the fact that Wiki tables tend to be relatively small themselves, is this the case?

o2) I am not convinced that cosine similarity is sufficient to guarantee good matches. Assuming that joins across tables in a RDB are exact, any discrepancy between the strings in the table would lead to missing joins. Is this avoided by using the Wikidata item ID? Was cosine similarity the only method that was studied? It would be good to add detail on the accuracy of this method (false positives and false negatives), and on the average overlap between tables in a dataset (e.g., Jaccard Containment [1]). I can imagine this would be an expensive step to perform over the entire database, however as exact joins are prevalent in practice it is important to have an idea of whether they can be performed properly on WikiDBs. Furthermore, while working exclusively on the column names may be more efficient, any information found in the columns themselves is lost.

o3) On the subject of column names, in the submission it is argued that the column names tend to repeat themselves and there is little context coming from the table itself: this is then used as justification for renaming the columns with GPT-4 for the later steps. However, this does cast doubt that the embeddings generated by SentenceBERT can be quite as effective if the name of the columns is not that informative. On the other hand, I also believe that the generated names would not be so useful in practice, as RDBs may very well feature column names that are automatically generated and that carry very little information on the actual content of the table, or no proper column names at all. Indeed, having names that have been automatically crafted might make the benchmark _easier_ than it would be with the original, uninformative names. It would be good to have a version of the data lake that does not include the GPT-generated table names: this would also allow to gauge how methods that look only at the content of the columns (rather than the names) fare in different scenarios (with and without informative column names). The "base-name" version may well be a set of schemas that replaces the new names, rather than a full copy of the dataset.

o4) There is not enough information about the computational resources required to work with the database, nor to pre-process the data. Please add the size of WikiDBs (and the other baselines) to the main body, and add information on the resources required to prepare the dataset, as this would be important for reproducibility and to better use the dataset in practice.

o5) One of the methods evaluated in the experimental section combines a LM with a GNN, without providing a justification for it. Moreover, no information of what the graph is supposed to be based on. Is the graph at the level of a database? How are tables modelled as graphs? Please add more detail on the reasoning behind the choice and on how the GNN infrastructure is implemented.

o6) The experimental section could have been more exhaustive. While the intent of the paper of creating an "LDM" is clear and well explained, the experiments used to justify it are not very compelling. I am not sure how useful it would be in practice to optimize the exact naming of columns in a database, rather than acting on the actual content of the database itself. A large dataset such as WikiDBs would be useful for various applications, not limited to training LLMs: for example, it could be used to benchmark unionability [2] or table discovery methods [1] or [3]. While running a new set of experiments on the entire dataset may be impractical, it would be interesting to see at least some degree of discussion and preliminary results on the subject, or observations on the feasibility of such applications.

[1] Fernandez, Raul Castro, et al. "Lazo: A cardinality-based method for coupled estimation of jaccard similarity and containment." _2019 IEEE 35th International Conference on Data Engineering (ICDE)_. IEEE, 2019.
[2] Khatiwada, Aamod, et al. "Santos: Relationship-based semantic table union search." _Proceedings of the ACM on Management of Data_ 1.1 (2023): 1-25.
[3] Zhu, Erkang, et al. "Josie: Overlap set similarity search for finding joinable tables in data lakes." _Proceedings of the 2019 International Conference on Management of Data_. 2019.

**Relation To Prior Work:**

Prior work considers various RDB benchmark alternatives. As gittables is considered among the baselines, it would be good to add datasets such as those prepared in [1], which include dumps of RDBs without a clear schema. Datasets like the one mentioned should provide more info on what a "realistic" DB looks like, although they may contain dirtier data and no schema.

[1] Khatiwada, Aamod, et al. "Santos: Relationship-based semantic table union search." _Proceedings of the ACM on Management of Data_ 1.1 (2023): 1-25.

**Summary And Contributions:**

The main contribution of the submission is WikiDBs, an open-souce corpus of 100k relational databases and 1.4M tables based on Wikidata, to be used as a foundation for Large Database Models. Detailed instructions on how to construct and extend the dataset are also provided, along with an experimental section that introduces potential use cases for LMs on the dataset.

---

> ### Author Rebuttal · Authors · 2024-08-16
>
> Part (1/4)
>
> We thank the reviewer for their very detailed, in-depth review and positive feedback, as well as the constructive suggestions for improvement. We will now address the comments one by one:
>
> > While the overall size of the dataset is large, tables are quite small as the average number of rows is in the range of few hundreds. Typical relational databases tend to involve a much larger number of records, which is not reflected in the current version of WikiDBs. I imagine this is due to the fact that Wiki tables tend to be relatively small themselves, is this the case?
>
> We agree that tables with more rows are an important direction for a corpus such as WikiDBs. Unfortunately, Wikidata does not provide more data. To maximize table sizes in WikiDBs, we do a full "profiling" of Wikidata, where we collect sizes every possible table (i.e., for every topic in Wikidata).  For our corpus, we then take the largest tables from each domain as starting tables for our construction process of databases and use the neighboring tables as additional tables.
>
> Unfortunately, this still does not lead to large tables overall, as Wikidata only provides a few hundred entries at maximum per topic.  When comparing the table sizes, however, to other corpora such as GitTables which also provide millions of tables (but only individual tables), we see that GitTables also does not provide more rows per table.  Overall, we believe that WikiDBs thus will provide value to the community since it is the first large-scale corpus with in total millions of tables connected as many databases and thus can foster research on tabular foundational models / large database models (LDMs).
>
> As a future route, we agree that extending WikiDBs and providing larger tables is an important future direction though. One interesting observation is that while Wikidata is very large, it is far from complete. As such, one potential avenue is to add more data by trying out data augmentation strategies on the databases in WikiDBs or use LLMs to synthesize additional data from world knowledge and include new rows for topics that are not (yet) present in Wikidata.
>
> >  I am not convinced that cosine similarity is sufficient to guarantee good matches. Assuming that joins across tables in a RDB are exact, any discrepancy between the strings in the table would lead to missing joins. Is this avoided by using the Wikidata item ID? Was cosine similarity the only method that was studied? It would be good to add detail on the accuracy of this method (false positives and false negatives), and on the average overlap between tables in a dataset (e.g., Jaccard Containment [1]).
>
> We think there is a misconception about how we used cosine similarity, which we try to clarify next: we do not use cosine similarity to find suitable tables that are joinable from WikiDBs. Instead, for constructing databases of connected tables we use the existing references in Wikidata as Wikidata is a document-oriented knowledge base that stores information on each document (Wikidata item, e.g., "New York") in a triple format (subject-predicate-object). Therefore, references between tables do not need to be constructed by finding joinable tables. Instead, with Wikidata we can use references from the Wikidata graph to construct foreign key tables.
>
> Re how we use cosine similarity:  When constructing databases for WikiDBs, we start with a starting table and construct connected tables as neighboring tables. However, there are typically multiple choices which connected tables can be included in a databases. To be more precise, all columns of a starting table which refer to other topics (i.e., all coluns which refer to other topics which have their own Wikidata page) are potential candidates for neighboring tables.
> For example, assume we have an *astronauts* table as starting table which has columns such as *employer*, *mission*, and *citizenship* that can be used as foreign keys to create a neighboring table. In order to select a neighboring table that we include in a database in WikiDBs, we aim to pick a foreign key that is "less generic" and "more closely related" to the start table  *astronauts*. To select the best suiting table as neighboring table from a set of candidates, we use cosine similarity between starting table name and all possible canidate tabbles. In particular, we use SentenceBert embeddings of the start table name and the possible candidate table name. In the example above, we get the highest similarity for the *mission* table (which is also the one we would select), while the *employer* and *citizenship* neighboring table names are more generic (i.e., less similar) and would thus not be selected.
>
> Re is this a good procedure? Clearly, while cosine similarity is a common choice for similarity on word embeddings, also other similarity metrics could be used.
> Also, this procedure is clearly a heuristic to select neighboring tables for incuding them in a database that are more domain-specific and less generic.
> Since we will open source our generation scripts for Wikidata, these are all interesting variations one can try out in future. However, as there is no direct way of measuring what is a better choice (i.e., which similarity measure, or more domain-specific or more generic neighnoring tables), this can be only evaluated based on downstream benchmarks (i.e., which data is more beneficial for pre-training an LDM and use it for realizing downstream tasks). This is what we want to foster with out research.

---

> > ### Author Rebuttal · Authors · 2024-08-16
> >
> > continued (Part 2/4):
> >
> > > On the subject of column names, in the submission it is argued that the column names tend to repeat themselves ...  Indeed, having names that have been automatically crafted might make the benchmark easier than it would be with the original, uninformative names.
> >
> > We would like to clarify that our motivation for renaming columns does not stem from a desire for having a cleaner dataset, but rather from the fact that the original column names derived from Wikidata are too uniform, i.e., our constructed dataset would be too "clean" and too "uniform" without this renaming.  By design, the properties in Wikidata, which we use for the column names, are not very diverse, as there are only in total about 11k properties (and thus 11k different possible column names) in Wikidata. However, our corpus contains about 70 million columns in total. Therefore, without renaming the original column names would repeat very often. By renaming columns from, e.g., "field of work" to "research field" or "description" of a person to "biography" based on the context, we get a more diverse dataset.  Overall, we have around 84k different column names after renaming, which is significantly more diverse than the 11k different column names available in Wikidata.
> > Finally, it is important to mention that in our dataset, we include both the renamed column names and the original column names (please see the *renaming.json* file included in each database folder), so that users can choose which set of column names to work with, depending on their use case.
> >
> > Re difficulty on LLM-renamed data: Our experiment in Section 5.3 (see Table 2 in our paper) show that the performance of using LLMs for predicting column names on an unseen test set using column names that have been renamed is lower than on columns that have not been renamed, indicating that the renaming makes the task more difficult as the data becomes more diverse. Important is also that the setup for renaming column names during data set construction and during predicting column names is different, as for the renaming we additionally use the original column name as input to the LLM while this information is not provided for the task.
> >
> > > On the other hand, I also believe that the generated names would not be so useful in practice, as RDBs may very well feature column names that are automatically generated and that carry very little information on the actual content of the table, or no proper column names at all._
> >
> > We agree with the reviewer that the characteristics of real-world databases can be quite diverse, especially in terms of data types, cell values or column names used, and not all of them are refelected in WikiDBs. We will discuss this limitation in the paper. As an example, enterprise data has different characteristics [1] (e.g., wide tables, numeric data, more cryptic column names, sparistiy of data).  However, some of these aspects can be addressed by filtering down the content to specific columns (e.g., only numerical columns) or by renaming column names or dropping cell values to make data more sparse. See also answer to reviewer Gs1a (3rd point). We will also include this discussion in the paper as an interesting future avenue.
> >
> > [1] Bodensohn, Jan-Micha, et al. "LLMs for Data Engineering on Enterprise Data". 2nd Workshop on Tabular Data Analysis (TaDA@VLDB'24). 2024
> >
> > > Indeed, having names that have been automatically crafted might make the benchmark easier than it would be with the original, uninformative names.
> >
> > First, we want to mention that the original column names of Wikidata are themselves informative (as discussed before).
> > Moreover, with the renaming we increase the variety of column names which indeed makes our benchmark tasks more difficult.
> > For example, for predicting missing column names, table names and cell values (our experiments in Table 2), we show that the performance on the original names is higher than on the renamed table and column names, indicating that the column names crafted by GPT-4 make the benchmark harder for these tasks. We attribute this to the fact that renaming makes the benchmark much more diverse, since in the original corpus, we only have 11k different column names, as they are standardized by Wikidata, while after renaming we have 84k different column names.
> >
> > >It would be good to have a version of the data ... that does not include the GPT-generated table names
> >
> > We agree with the reviewer that both the original and the renamed names should be included in the corpus, and we have already done so. Please see the *renaming.json* file included in the database folders in our supplementary material, which contains the original names (left) and the renamed version (right).  When we release the corpus, we will include detailed documentation of what each database instance consists of; currently, this information is included in our datasheet.

---

> > > ### Author Rebuttal · Authors · 2024-08-16
> > >
> > > continued (Part 3/4):
> > >
> > > > There is not enough information about the computational resources required to work with the database, nor to pre-process the data. Please add the size of WikiDBs (and the other baselines) to the main body, and add information on the resources required to prepare the dataset, as this would be important for reproducibility and to better use the dataset in practice.
> > >
> > > We agree with the reviewer that the current documentation lacks information about the required computing resources for creating the corpus and statistics of WikiDBs. We will add the size of WikiDBs to the paper and the repository, and in the repository we will add information about all required resources for the corpus construction in case the generator should be rerun.
> > >
> > > Reuirements on disk space: For the setup around 50GB of disk space are necessary; for creating databases, on average around 5MB are necessary for each database. Our scripts are scalable and the number of worker processes for creating databases in parallel can be specified in our configuration file. On an Intel(R) Xeon(R) Gold 5120 CPU @ 2.20GHz we observe the following resource consumption:
> > >
> > > Reuirements on CPU and RAM: 1 core and ~25GiB per worker process. Each worker process creates approximately 20 databases per hour. We estimate a total of 5000 CPU-hours for 100.000 databases, which can be evenly divided among a chosen number of worker processes.
> > >
> > > > One of the methods evaluated in the experimental section combines a LM with a GNN, without providing a justification for it. Moreover, no information of what the graph is supposed to be based on. Is the graph at the level of a database? How are tables modelled as graphs? Please add more detail on the reasoning behind the choice and on how the GNN infrastructure is implemented.
> > >
> > > We agree with the reviewer that the justification for combining language models and GNNs is currently missing and we will add it to our experiments section.
> > >
> > > In line with related work such as TaBERT and RPT, the LM receives tables with linearized table rows as an input. Our experiments (Table 2) show that even when scaling up the language model parameters, it is not sufficient to achieve high performance on the tasks of predicting missing values, table and column names. Therefore, we believe that architectural improvements are necessary to allow models to incorporate more context from the table in a more structure manner and also to allow including related tables in a database in the input.
> > >
> > > To provide more context and explicitly represent the structure of relational databases, we model tables and databases as graphs. In the graph, table cells, columns, and the table name are represented by nodes, edges explicitly model the connections between elements in a table, as well as the foreign key relationships between tables. In our LM+GNN model, we use the LM encoder to create initial encodings for each graph node, which are updated with information from neighboring nodes through GNN message passing (i.e., we propagate information along the  table structure). Finally, we use the LM decoder to generate the masked value (see [2] for more details).
> > >
> > > [2] Liane Vogel, Benjamin Hilprecht, and Carsten Binnig. 2022. Towards Foundation Models for Relational Databases [Vision Paper]. Table Representation Learning Workshop at NeurIPS 2022 (2022)
> > >
> > > > The experimental section could have been more exhaustive. While the intent of the paper of creating an "LDM" is clear and well explained, the experiments used to justify it are not very compelling. I am not sure how useful it would be in practice to optimize the exact naming of columns in a database, rather than acting on the actual content of the database itself. A large dataset such as WikiDBs would be useful for various applications, not limited to training LLMs: for example, it could be used to benchmark unionability [2] or table discovery methods [1] or [3]. While running a new set of experiments on the entire dataset may be impractical, it would be interesting to see at least some degree of discussion and preliminary results on the subject, or observations on the feasibility of such applications.
> > >
> > > We agree that using the corpus for detecting joinable and unionable tables is an interesting direction. As mentioned before already, the main purpose of the paper is to provide a corpus to enable research on tabular foundational models / LDMs that are trained on multiple connected tables. For the camera-ready paper, we will add a section about potential downstream tasks of such a model and discuss how models for these downstream tasks can be trained and tested based on our corpus.
> > >
> > > We also aim to add at least one more task to our evaluation (e.g., semantic column type annotation) which shows how WikiDBs can be used in conjunction with other corpora to test the benefit of having a pre-trained large database model (LDM). The idea is to show that we can realize a classification model for semantic type annotation with low training overhead (e.g., a small training corpus) compared to learning a semantic type annotation model from scratch. We will provide the results also here once available.
> > >
> > > > Prior work considers various RDB benchmark alternatives. As gittables is considered among the baselines, it would be good to add datasets such as those prepared in [1], which include dumps of RDBs without a clear schema. Datasets like the one mentioned should provide more info on what a "realistic" DB looks like, although they may contain dirtier data and no schema.
> > >
> > > Currently, for pre-training foundation models for relational databases, we focus on corpora where databases with a clear schema are given to enable the pre-training as outlined in our experimental section. However, for downstream applications which need to deal with real-world dirty data, datasets like Santos are highly relevant. We will discuss this dataset in the paper as well and include a reference.

---

> > > > ### Author Rebuttal · Authors · 2024-08-16
> > > >
> > > > continued (Part 4/4):
> > > >
> > > > > It is also unclear where the dataset is stored at all.
> > > >
> > > > The preliminary 20k version of the dataset in the supplementary material is accessible for the reviewers on google drive (please refer to this [link](https://drive.google.com/drive/folders/1wMRFro0ydQghmYeavBaBv_IUsPobT_JK?usp=sharing) which is also provided in *supplementary.pdf*). We plan to publish the final version of WikiDBs on Zenodo as well as on the HuggingfaceDataset Hub, as described in our datasheet.
> > > >
> > > > > Please consider using parquet as a storage format, rather than CSV. Parquet files are compressed and store the schema used by the tables, whereas CSV files occupy far more space and do not retain information on the table schema.
> > > >
> > > > We thank the reviewer for the suggestion. Currently we provide CSV files and store the table schema in an additional JSON file such that they can be seperately used. For the final format, we will in addition provide parquet files.
> > > >
> > > > -------
> > > >
> > > > We hope the answers above helps with the main questions outlined by the reviewer and we are more than happy to answer any further questions.

---

> > > > ### Comment · Reviewer_nHYv · 2024-08-19
> > > >
> > > > Thanks for the responses, they address well my concerns.
> > > >
> > > > I understand that space is limited, but I believe that at least some additional detail on the GNN should be provided in the main body, and that the requirements to use (not generating) the benchmark should also be provided in the body. Further detail may go in the appendix.
> > > >
> > > > Expanding the future work section with a more expansive discussion of future work would also address more of the points I raised.

---

> > > ### Comment · Reviewer_nHYv · 2024-08-19
> > >
> > > Thanks for the response.
> > >
> > > In this instance, I disagree in principle with the stance of the authors on the subject of column naming.
> > >
> > > The point of generating new names for the columns makes sense if the objective is to benchmark how accurate a system is at guessing the exact name of the column: it is not surprising that it is easier to guess the original WikiData column names, rather than the newly generated column names.
> > >
> > > I believe this mismatch in opinion is due to the fact that the authors are focusing on the specific problem of benchmarking LLMs, a task that would definitely be made harder by changing the original names to names that are not likely to be found in the training set of said LLMs, and I agree that this approach would be effective in this instance. However, this dataset could also be used to test alternative methods and scenarios, such as those in corporate DMBS where there was no effort put into renaming columns to have a meaningful name (see my comment on benchmarking alternative tasks).
> > >
> > > I personally believe that this work could be applied with success for more tasks than the authors suggest, and that the original column names would be very effective at "tripping up" methods that rely exclusively on the column names to perform schema matching (for example), when compared to other methods that instead consider the content of the tables as well as their schema. Or, maybe I am wrong and indeed the new names are harder even in other conditions! The point is that having both versions would only bring benefits as different users may choose the variant that best fits their scenario.
> > >
> > > With this comment I do not want to diminish the work done by the authors, rather I want to highlight that the benchmark could have a larger reach by providing both versions. As the authors did make both versions public (which I appreciate), my only comment is to add some consideration on the use of the benchmark for further work in different scenarios.

---

> > ### Author Rebuttal · Authors · 2024-08-19
> >
> > > I would still advise the authors to rephrase this section in the paper to make sure that such a misconception is less likely to occur with other readers.
> >
> > Thanks for the response. We agree with the reviewer and will discuss the limitation of the table sizes in our paper. We will also rephrase the section where we explain how cosine similarity is used based on the explanations above.

---

> ### Comment · Reviewer_nHYv · 2024-08-19
>
> Thanks for the response.
>
> I did expect the size of wiki tables to be problematic: while the specific characteristics of the generated tables do not reduce the validity of this work, it is still a limitation that should be mentioned in the text.
>
> I have indeed misunderstood how cosine similarity is used in the generation process. I agree that using cosine similarity to choose "related tables" while relying on the triplets to execute the actual join is a valid strategy, and have no issues with the current application.
>
> I would still advise the authors to rephrase this section in the paper to make sure that such a misconception is less likely to occur with other readers.

---

> ### Comment · Reviewer_nHYv · 2024-08-19
>
> I thank the authors for addressing my concerns. I appreciate the effort put into explaining the less-clear sections of the work and the publication of both versions of the dataset.
>
> Personally, I think this benchmark could have a large impact on both training LLMs, and further data lake-oriented research problems and I thank the authors for the effort put into building it.

---

> > ### Author Rebuttal · Authors · 2024-08-19
> >
> > Thanks for the responses! We appreciate the reviewer's suggestions for improving our paper and corpus.
> >
> > > As the authors did make both versions public (which I appreciate), my only comment is to add some consideration on the use of the benchmark for further work in different scenarios.
> >
> > Together with the suggestions from Reviewers Gs1a, we will extend our future work section to cover more different scenarios, such as data lakes and data discovery tasks.
> >
> > > at least some additional detail on the GNN should be provided in the main body, and that the requirements to use (not generating) the benchmark should also be provided in the body
> >
> > We agree with the reviewer and will add details on the GNN to the paper. We will also add the resources necessary to work with our corpus to the main body of the paper. We already added the resources for reproducing the benchmark to our repository readme.

---

### Official Review · Reviewer_Gs1a · 2024-07-19
**Very useful resource, technically strong, and well-written paper**

**Rating:** 9
**Confidence:** 5
**Correctness:** I did not spot any incorrect statemen…
**Clarity:** The paper is well-written.

**Review:**

The work is well-motivated, the construction methodology that includes the use of LLMs for refinements is very reasonable, and the preliminary experiments show promising results.

The work is the first of its kind and has the potential to make a good impact on research in this space.

One place I see room for improvement is the experiments. It would have been nice to show how training or fine-tuning with WikiDBs can improve a model for a task over another dataset. For example, you could do similar base-T5 vs fine-tuned T5 experiments but over a SemTab column type annotation dataset and task, showing that if you fine-tune the model using WikiDBs, you get better results than the base model. Even better would be training/fine-tuning with one of SemTab datasets as well, and showing that training on WikiDBs works better. That to me would make this work much stronger.

It is great that the data generator is also publicly released, in addition to the dataset, which allows custom dataset generations. Another room for improvement could be creating datasets that are more realistic for certain tasks. For example, create tables with full numerical values, which are common in many real settings particularly when the goal is learning, e.g. classification.

**Strengths:**

- Potential for great impact
- Good design of data construction process
- Results showing the application in table completion tasks

**Additional Feedback:**

It would be great to form a community that could work on improvements of the data generation process. Perhaps this can be done in the form of a task or competition/challenge at a workshop or conference, such as SemTab, TRL or TaDA workshops.

**Documentation:**

The dataset has a very good documentation.

**Ethics:**

I do not find any ethical concerns about this work.

**Limitations:**

The limitations are discussed well. I suggest discussing or addressing the limitation that some relational databases may look very different from in terms of characteristics (e.g., having all numeric values, lots of timestamps, very large tables, etc.) and WikiDBs may not reflect that.

**Opportunities For Improvement:**

As stated above, I see two opportunities for improvement. One is generating a more diverse collection of relational databases, or providing guidelines on how such databases can be constructed or what it takes to improve your generator to do so. And example is databases of mostly numerical contents.

Another opportunity for improvement is proving the usefulness of the dataset for training models that outperform state-of-the-art solutions on another task/dataset, such as the SemTab tasks or data discovery tasks, such as those in LakeBench: https://arxiv.org/abs/2307.04217

**Relation To Prior Work:**

Related work discussion is very reasonable. You may want to discuss LakeBench and data discovery tasks that can benefit from your work: https://arxiv.org/abs/2307.04217

**Summary And Contributions:**

The paper describes WikiDBs, which is a large corpus of relational databases carefully constructed from Wikidata. A strong motivation is presented for the need for this dataset, along with a good discussion or related work. Then the construction methodology is described in detail, including the use of LLMs to make the resulting databases more realistic. Preliminary experiments are performed on three table completion tasks.

---

> ### Author Rebuttal · Authors · 2024-08-16
>
> We thank the reviewer for their in-depth review and their very motivating and positive feedback, as well as the constructive suggestions for improvement. We will now address the comments one by one:
>
> > One place I see room for improvement is the experiments. It would have been nice to show how training or fine-tuning with WikiDBs can improve a model for a task over another dataset.
>
> We agree with very much with the reviewer.
>
> Re focus of the paper: The main focus of this paper is to build a corpus that we hope will support more extensive research on how to realize foundational database models which can be used for fine-tuning since this is still a very much open question. However, we are more than happy to include a first experiment with our own model architecture in the paper that shows the benefits of having a pre-trained task-independent model vs. training a task-dependent model from scratch. We initially did not include such a experiment into the paper due to space constraints and our focus on the corpus itself. However, as the final version allows one additional page, we will include such an experiment.
> We are currently running an experiment for column type annotation using a setup similar to [1,2] and will make the results available as soon as we have them and then include them in the paper.
>
> [1] Suhara, Yoshihiko, et al. "Annotating columns with pre-trained language models." Proceedings of the 2022 International Conference on Management of Data. 2022.
> [2] Hulsebos, Madelon, et al. "Sherlock: A deep learning approach to semantic data type detection." Proceedings of the 25th ACM SIGKDD International Conference on Knowledge Discovery & Data Mining. 2019.
>
> > One is generating a more diverse collection of relational databases, or providing guidelines on how such databases can be constructed or what it takes to improve your generator to do so. An example is databases of mostly numerical contents.
>
> We agree and think that creating subsets with specific characteristics such as mostly numerical content is a very interesting avenue of future work. In fact, our generator can be adapted to do so. Our generator already allows for customization in terms of many parameters such as number of tables and number of columns oper tabler etc.  In the [code repository](https://github.com/DataManagementLab/wikidbs-public) of our generator, we will include documentation on how to extend the generator to provide further parameters (e.g., for filtering out specific contents such as columns from certain types).
>
> > I suggest discussing or addressing the limitation that some relational databases may look very different from in terms of characteristics (e.g., having all numeric values, lots of timestamps, very large tables, etc.) and WikiDBs may not reflect that.
>
> We agree with the reviewer that the characteristics of real-world databases can be quite diverse, especially in terms of data types, cell values or column names used, and not all of them are refelected in WikiDBs. We will discuss this limitation in the paper. As an example, enterprise data has different characteristics (e.g., wide tables, numeric data, more cryptic column names, sparistiy of data).
> However, as outlined above, some of these aspects can be addressed by filtering down the content to specific columns (e.g., only numerical columns) or by renaming column names or dropping cell values to make data more sparse. We will include this discussion in the paper as an interesting future avenue.
>
> > You may want to discuss LakeBench and data discovery tasks that can benefit from your work: [https://arxiv.org/abs/2307.04217](https://arxiv.org/abs/2307.04217)
>
> We thank the reviewer for the suggestion and will include the dataset in the part of our paper where we also discuss other (related) datasets.
>
> > It would be great to form a community that could work on improvements of the data generation process. Perhaps this can be done in the form of a task or competition/challenge at a workshop or conference, such as SemTab, TRL or TaDA workshops.
>
> This is indeed a great suggestion that we also have thought of as a next step. The discussion about benchmark tasks as suggested by reviewer aNTS is a good starting point for this.
>
> -------
>
> We hope the answers above helps with the main questions outlined by the reviewer and we are more than happy to answer any further questions.

---

### Official Review · Reviewer_HLok · 2024-07-21
**Useful corpus, unclear that the empirical evaluation is sufficiently convincing though**

**Rating:** 7
**Confidence:** 5
**Clarity:** Yes, the paper is well written.

**Review:**

This paper creates a corpus of 100K relational databases with 1.4M tables, and an average of about 3 tables per database from Wikidata, which seems like a very valuable corpus.  In particular the authors make the valid point that no such corpus exists compared to others in terms of scale.  Connectivity between tables within a database is extremely useful for tasks such as representational learning of tables, particularly if one wants to teach neural systems the notion of database constraints.  The authors argue that "We hope that our corpus fosters more research on models for table representation learning that can take data from multiple connected tables into account.", and this is indeed a very interesting avenue to pursue.  However, the current set of experiments to help showcase the value of the corpus is a bit weak.

**Strengths:**

1.  Careful creation of a large corpus of databases from Wikidata, which seems to be better than most of the other datasets in the literature in terms of scale (100K databases, 1.4M tables).
2.  Clarity in writing.

**Additional Feedback:**

Please answer my questions about usefulness of corpus from the empirical work.

**Correctness:**

See above - I do think the corpus is very useful but I do not believe that the current experiments make the case empirically.

**Documentation:**

Yes

**Limitations:**

1. The initial experiments seemed weak compared to the development of the corpus.  What was the rationale for the choice of tasks - finding missing values, column names, or table names - why would these be impacted by the presence of multiple relational tables?
2.  What was the reason to choose fine tuning with multiple versions of T5 or BART?  Why these specific models?
3.  What would happen if the individual tables were given to the models - would they perform worse?  This latter test seems crucial to make the point that representational learning will benefit from having multiple tables in a dataset?

**Opportunities For Improvement:**

1. The initial experiments seemed weak compared to the development of the corpus.  What was the rationale for the choice of tasks - finding missing values, column names, or table names - why would these be impacted by the presence of multiple relational tables?
2.  What was the reason to choose fine tuning with multiple versions of T5 or BART?  Why these specific models?
3.  What would happen if the individual tables were given to the models - would they perform worse?  This latter test seems crucial to make the point that representational learning will benefit from having multiple tables in a dataset?

**Relation To Prior Work:**

Yes

**Summary And Contributions:**

This paper creates a corpus of 100K relational databases with 1.4M tables, and an average of about 3 tables per database from Wikidata, which seems like a very valuable corpus.  In particular the authors make the valid point that no such corpus exists compared to others in terms of scale.  Connectivity between tables within a database is extremely useful for tasks such as representational learning of tables, particularly if one wants to teach neural systems the notion of database constraints.  The authors argue that "We hope that our corpus fosters more research on models for table representation learning that can take data from multiple connected tables into account.", and this is indeed a very interesting avenue to pursue.  However, the current set of experiments to help showcase the value of the corpus is a bit weak.

---

> ### Author Rebuttal · Authors · 2024-08-16
>
> Part (1/2)
>
> We thank the reviewer for the valuable comments. Below, we address each of the questions individually. Most importantly, we will revise the experiments section of the paper to provide more detail on the motivation for the choice and design of the experiments.
>
> >The initial experiments seemed weak compared to the development of the corpus. What was the rationale for the choice of tasks - finding missing values, column names, or table names - why would these be impacted by the presence of multiple relational tables?
>
> Thanks for the comment. We agree that the goal of a tabular foundational model is to support many more downstream tasks and we see our corpus as an important starting point for enabling the training and development of LDMs, which can then be used in many different downstream applications similar to how LLMs are used today on text-based tasks.
>
> Re tasks used in the evaluation: Missing value imputation and prediction of column and table names are standard data engineering tasks and have been used in the literature [3]. Moreover, we have chosen those tasks as they require an understanding of table structures at different granularities: While cell value imputation requires context from data in the same row and column, table name prediction can benefit from the full table as well as from connected tables in a database.
>
> Re why neighboring tables might help for these tasks? In general, we believe that context can in principle help in all the before mentioned tasks but the usefulness depends very much on the concrete data set. For example, for finding a table name of a table with personal data depends on the context. If the neighboring table is a movie table, then the table name might be rather actor than sports player where the neighboring table contains a list of sports teams. However, how different tabular data understanding tasks are impacted by the presence of more context in form of multiple relational tables is exactly the line of research that we want to enable with our corpus. For the camera-ready version of the paper, we plan an additional experiment where we provide a first analysis of whether including more context from related tables improves the performance of the imputation task or not. We post the results as soon as they are available.
>
> Re use of the corpus also for single tables: while we constructed the WikiDBs corpus to enable research on tabular foundational models which take multiple tables of a database into account, the corpus can also be used for research on tabular foundational models for single tables.
>
> We are going to clarify these points in Section 1 (Introduction) and Section 5 (Initial experiments) of our paper.
>
> > What was the reason to choose fine tuning with multiple versions of T5 or BART? Why these specific models?
>
> The focus of this paper is to establish a corpus that can be used as a starting point for research on tabular foundational models or large database models as we call them in the paper (LDMs) which are pre-trained on a large variety of databases with multiple tables. However, while there are first suggestions of how tabular foundational models / LDMs can look like, the question about the design of such models and what model sizes are needed is not (yet) understood very well and we thus believe that a corpus such as ours will signifcantly help to push this area forward.
>
> Re choice of models: In the paper, we have chosen model architectures based on related work in tabular represenation learning. TaBERT [1] has trained BERT and RPT[3] has trained BART on single tables. We have chosen BART and T5 because they are encoder-decoder language models, and as such enable to solve generation based downstream tasks, such as generating unseen missing values. We therefore think that they are a good starting point for tabular foundation models.
>
> Re different versions of T5: As discussed before, it is not yet well understood what sizes in terms of number of parameters are needed to build tabular foundation models / LDMs.  Thus, we decided to experiment with different model sizes and analyze the effects on accuracy. We included the T5 models because they come in sizes up to 11 billion parameters, while the largest BART model has only about 406 million parameters. However, we also want to emphasize that model size is only one open question. Other questions are the model architeture itself as it is shown by other papers which study different variants of how to enhance LLMs [4,5,6] or they combine LLMs with other model architectures such as GNNs [7,8] (which includes our own prior work [9]). To show this effect and highlight that the corpus can enable research on designing and evaluating model architecture and model sizes for tabular foundational models/LDMs, we included T5 in different sizes as a first strawman experiment. While we see that increasing the model parameters increases the performance on all tested tasks, there are diminishing returns and even with the largest model the performance scores are all below 80% (see table 2).
>
> We agree with the reviewer that motivation of why we chose T5 and BART as starting points was not clear in the experiment section, and we will add the information based on the text above to the paper.

---

> > ### Author Rebuttal · Authors · 2024-08-16
> >
> > continued (Part 2/2):
> >
> > > What would happen if the individual tables were given to the models - would they perform worse? This latter test seems crucial to make the point that representational learning will benefit from having multiple tables in a dataset?
> >
> > Re are multiple tables benefitial: As mentioned before, we believe context from neighoring tables is helpful but it depends on the data set and task. Currently, the quesion on how tabular foundational models should be designed is at the very beginning and we constructed our corpus to enable research on the architecture and training procedure for such tabular foundational models. This research also includes the question on when, how, and to which extent context from neighboring tables is more helpful than only using individual tables.
> >
> > Re experiment showing the benefit of multiple tables: In line with related work such as TaBERT and RPT, the LM in our experiment in table 2 receive linearized table rows from a single table as an input. We see that this context is not sufficient to predict (unseen) table names, column names, and cell values with very high performance.
> > To show the benefits of multiple tables, in the camera-ready version of the paper, we aim to include a lightweight experiment for the benefit of adding multiple tables.
> > We performed a preliminary experiment on table name prediction, investigating how much the context of neighboring tables helps. We prompted GPT-4o-mini (gpt-4o-mini-2024-07-18) to predict a missing table name from 250 randomly selected databases out of WikiDBs, once given only the table schema of the single table and once given additionally the table names and schemas from the directly connected tables from the database. We observe that the exact match score increases from 0.084 to 0.128 when given the additional context.
> >
> > -------
> >
> > We hope the answers above helps with the main questions outlined by the reviewer and we are more than happy to answer any further questions.
> >
> > -------
> >
> > [1] Yin, Pengcheng, et al. "TaBERT: Pretraining for Joint Understanding of Textual and Tabular Data." Proceedings of the 58th Annual Meeting of the Association for Computational Linguistics. 2020.
> > [2] Deng, Xiang, et al. "Turl: Table understanding through representation learning." ACM SIGMOD Record 51.1 (2022): 33-40.
> > [3] Tang, Nan, et al. "RPT: relational pre-trained transformer is almost all you need towards democratizing data preparation." Proceedings of the VLDB Endowment 14.8 (2021): 1254-1261.
> > [4] Chen, Pei, et al. "HYTREL: Hypergraph-enhanced tabular data representation learning." Advances in Neural Information Processing Systems 36 (2024).
> > [5] Li, Peng, et al. "Table-GPT: Table Fine-tuned GPT for Diverse Table Tasks." Proceedings of the ACM on Management of Data 2.3 (2024): 1-28.
> > [6] Badaro, Gilbert, Mohammed Saeed, and Paolo Papotti. "Transformers for tabular data representation: A survey of models and applications." Transactions of the Association for Computational Linguistics 11 (2023): 227-249.
> > [7] Yasunaga, Michihiro, et al. "QA-GNN: Reasoning with Language Models and Knowledge Graphs for Question Answering." Proceedings of the 2021 Conference of the North American Chapter of the Association for Computational Linguistics: Human Language Technologies. 2021.
> > [8] Yasunaga, Michihiro, et al. "Deep bidirectional language-knowledge graph pretraining." Advances in Neural Information Processing Systems 35 (2022): 37309-37323.
> > [9] Liane Vogel, Benjamin Hilprecht, and Carsten Binnig. 2022. Towards Foundation Models for Relational Databases [Vision Paper]. Table Representation Learning Workshop at NeurIPS 2022 (2022)

---

> > ### Comment · Reviewer_HLok · 2024-08-16
> >
> > Thank you for your response.  Your points about T5/BART make sense.  I really do like the dataset and think it's useful - however, I feel that the experiments should have supported and demonstrated its specific value.  Adding the experiments proposed definitely would make things better, but without the experiments conducted and added to the paper, it is still difficult to evaluate in general the benchmark's usefulness - I understand that in principle this should help, but it would greatly strengthen the paper if the experiments were conducted and we had a sense of the results?

---

> > ### Author Rebuttal · Authors · 2024-08-27
> >
> > Thanks for your reply. We have run two additional experiments to address your comments and demonstrate the use of our corpus. Please see our response to our general "Rebuttal by Authors" at the top of the page for a detailed explanation.

---

### Official Review · Reviewer_aNts · 2024-07-26
**A substantial corpus of 100K relational databases aimed to foster research in table representation learning.**

**Rating:** 8
**Confidence:** 4
**Correctness:** Overall the statements presented in t…

**Review:**

**Quality:** The dataset construction methodology is overall well-designed and seems appropriate for the problem at hand. The produced data corpus is very large and the size seems adequate for table representation learning tasks.

**Clarity:** The paper is well-structured with a clear and easy-to-follow writing style. I wasn't able to find any noteworthy issues.

**Originality:** The authors mostly leverage well-established methods for building their dataset. There do not seem to be many such datasets out there, especially ones of this size, which makes this a noteworthy contribution.

**Significance:** Datasets like this have the potential to be widely used by the table representation learning community.

**Strengths:**

**(S1)** The dataset construction methodology is overall well-designed and seems appropriate for the problem at hand. The produced data corpus is very large and the size seems adequate for table representation learning tasks. Datasets like this have the potential to be widely used by the table representation learning community.

**(S2)** The paper is well-structured with a clear and easy-to-follow writing style.

**(S3)** The presented methodology can be used as a blueprint for constructing other datasets in the future.

**Additional Feedback:**

**(A1)** The abstract contains a sentence that caught my eye: "The corpus is based on Wikidata and follows the characteristics of real-world databases." I would try to slightly tone down this claim since one could find various ways to define "characteristics of real-world databases" and could argue that the specific characteristics that the authors look at are not sufficient. I would instead rephrase this to say that the authors made their best effort to follow some characteristics of real-world data that they deemed necessary/relevant.

**Clarity:**

The paper is well-structured with a clear and easy-to-follow writing style. I wasn't able to find any noteworthy issues.

**Documentation:**

The supplementary material contains a relatively comprehensive datasheet and a croissant description.

**Ethics:**

I was not able to identify any ethical concerns.

**Limitations:**

The authors include a brief limitations section where they discuss some interesting limitations of their work (e.g. the overrepresentation of certain types of entities in Wikidata).

**Opportunities For Improvement:**

**(O1)** This might be my personal bias, but I have slight reservations about using GPT for renaming columns. I understand the desire to have a cleaner dataset. On the flip side, many real-world datasets are quite messy, so by cleaning them up, the authors might be removing this "naturally" occurring noise. On top of that, the authors showcase a column name prediction method that predicts column names that were in fact derived from LLMs. Hence, one could raise questions about the real-world value of predicting such column names. I'm not sure what the right course of action is, and the authors do provide a brief discussion about their motivation to use an LLM for this, but I am raising this as a concern and think it might be seen as a potential limitation of this work.

**(O2)** (*non-fundamental*) This work is framed mainly as a dataset to be used for various table representation learning tasks, which is valid and valuable. That said, the authors could have also included some clearly defined benchmark tasks that could motivate the community to build models to solve them. One can see the three evaluation tasks as examples of benchmark challenges, but based on my reading of the paper, they weren't framed as such.

**Relation To Prior Work:**

The authors present many instances of existing related work. They also compare the dataset statistics of the dataset they produced with those of existing benchmarks.

**Summary And Contributions:**

The authors introduce a benchmark containing 100K relational databases with multiple tables that are linked together with foreign key relationships. Unlike existing benchmarks containing tables crawled from the web, the intention of this benchmark comprised of multi-table databases as instances. The authors build their benchmark by extracting information from Wikidata. Since Wikidata has a graph structure where entities have properties and links to other entities, the authors select meaningful entities as starting points to build the initial tables. After that, they use a mechanism they designed to follow the links in order to build out the linked tables. After describing the dataset construction methodology, the authors present various statistics of their dataset and compare them with other relevant datasets. Finally, the authors demonstrate how their dataset can be used to build table representation learning methods for missing value imputation, as well as column and table name prediction.

---

> ### Author Rebuttal · Authors · 2024-08-16
>
> Part (1/2)
>
> We thank the reviewer for their in-depth review and their positive feedback, as well as their constructive suggestions for improvement. We address the comments below one by one:
>
> > I have slight reservations about using GPT for renaming columns. I understand the desire to have a cleaner dataset. On the flip side, many real-world datasets are quite messy, so by cleaning them up, the authors might be removing this "naturally" occurring noise.
>
> We would like to clarify that our motivation for renaming columns does not stem from a desire for having a cleaner dataset,  but rather from the fact that the original column names derived from Wikidata are too uniform, i.e., our constructed dataset would be too "clean" and too "uniform" without this renaming.  By design, the properties in Wikidata, which we use for the column names, are not very diverse, as there are only in total about 11k properties (and thus 11k different possible column names) in Wikidata. However, our corpus contains about 70 million columns in total. Therefore, without renaming the original column names would repeat very often. By renaming columns from, e.g., "field of work" to "research field" or "description" of a person to "biography" based on the context, we get a more diverse dataset.  Overall, we have around 84k different column names after renaming, which is significantly more diverse than the 11k different column names available in Wikidata.
> Finally, it is important to mention that in our dataset, we include both the renamed column names and the original column names (please see the *renaming.json* file included in each database folder), so that users can choose which set of column names to work with, depending on their use case.
>
> > the authors showcase a column name prediction method that predicts column names that were in fact derived from LLMs. Hence, one could raise questions about the real-world value of predicting such column names
>
> Re general importance of task: We have chosen this task since predicting column names is a well-established task in the area of data engineering and it can be used to test how well models understand data in table columns as well as the context (i.e., other columns of a table).  For example, deriving column names is important when schema information must be derived from data (e.g., a CSV file without a header). This task is also closely related to other established tasks such as annotating semantic column types to table data for data search and discovery [1, 2].
>
> Re difficulty on LLM-renamed data: Our experiment in Section 5.3 (see Table 2 in our paper) show that the performance of using LLMs for predicting column names on an unseen test set using column names that have been renamed is lower than on columns that have not been renamed, indicating that the renaming makes the task more difficult as the data becomes more diverse. Important is also that the setup for renaming column names during data set construction and during predicting column names is different, as for the renaming we additionally use the original column name as input to the LLM while this information is not provided for the task.
>
> We will add this information to the paper.
>
> > the authors could have also included some clearly defined benchmark tasks that could motivate the community to build models to solve them
>
> We agree. For the camera-ready submission, we will add a section on which data engineering tasks we think can be supported by pre-trained tabular foundational models based on WikiDBs. We will also discuss how the different benchmarking tasks stress different parts of tabular foundation models: For example, while tasks like column type annotation require column understanding, tasks like entity matching and deduplication require to understand and compare rows. On schema level, tasks like schema matching or unionability detection are testing the table-level understanding of models.
>
> In addition to the discussion which tasks can be addressed by pre-trained tabular foundational models, we will also add a discussion for which benchmark tasks WikiDBs data can be used directly for testing. For example, for some of the tasks mentioned before like deduplication, WikiDBs might be easy to adapt while for others (e.g., column type annotation) additional existing data sets can be used (while WikiDBs is still being used for pre-training) or additional efforts are reqiured for creating ground truth annotations on WikiDBs to support these tasks.
>
> Finally, as part of our future work, we plan to see how well tabular foundational models that are pre-trained on WikiDBs can be used to solve a broad range of data engineering benchmark tasks. For the camera-ready paper (which has one extra page), we will include the results for one additional task (e.g., semantic type annotation) and show the benefits of using a pre-trained tabular foundational model vs. training a model. We are currently running such an experiment for column type annotation using a setup similar to [1,2] and will make the results available as soon as we have them and then include them in the paper.

---

> > ### Author Rebuttal · Authors · 2024-08-16
> >
> > continued (Part 2/2):
> >
> > > The abstract contains a sentence that caught my eye: "The corpus is based on Wikidata and follows the characteristics of real-world databases." I would try to slightly tone down this claim since one could find various ways to define "characteristics of real-world databases"
> >
> > We agree with the reviewer and will rephrase the sentence in the abstract to tone down the claim: "The corpus is based on Wikidata and aims to follow certain characteristics of real-world databases."
> >
> > -------
> >
> > We hope the answers above helps with the main questions outlined by the reviewer and we are more than happy to answer any further questions.
> >
> > -------
> >
> > [1] Suhara, Yoshihiko, et al. "Annotating columns with pre-trained language models." Proceedings of the 2022 International Conference on Management of Data. 2022.
> > [2] Hulsebos, Madelon, et al. "Sherlock: A deep learning approach to semantic data type detection." Proceedings of the 25th ACM SIGKDD International Conference on Knowledge Discovery & Data Mining. 2019.

---

> > > ### Comment · Reviewer_aNts · 2024-08-19
> > > **Response to the Rebuttal**
> > >
> > > Thank you for responding to the comments I raised. I will keep my current score in support of having this paper accepted.
> > >
> > > (GPT for renaming columns) I think we might have a different definition of what it means for column names to be clean. I would argue that the main goal of a benchmark dataset like this is to enable us to predict how well a model trained/evaluated on it will perform in real-world datasets. Hence, the closer we are to a real-world dataset the better (in terms of data distribution). I understand the motivation for renaming columns and I appreciate the arguments. At the end of the day, the fact that you're sharing both versions of the names is definitely a way to resolve this discussion, and we can let the people choose what they prefer.
> > >
> > > (predicting column names that were derived from LLMs) Thank you for the clarification, I missed that in the paper.
> > >
> > > (clearly defined benchmark tasks) Makes sense, thanks!
> > >
> > > (abstract) Cool, this sounds better, thanks!

---

> > > > ### Author Rebuttal · Authors · 2024-08-20
> > > >
> > > > Thanks for the response and the positive feedback! We will add the clarification about predicting column names that were derived from LLMs to our paper. We will also clearly state that there are two sets of column names available to choose from.

---

### Author Rebuttal · Authors · 2024-08-16

We thank all reviewers for their in-depth reviews and appreciate their valuable feedback to improve our corpus and our paper.

The main focus of this paper is to build a corpus that we hope will support more extensive research on how to realize large database models, and we appreciate that the reviewers acknowledge the benefit that our corpus can bring for the community.
Based on the feedback, we will add a section about potential downstream tasks of large database models and discuss how downstream tasks can be trained and tested based on our corpus.
We will also add more detailed information to our paper about our choice of models and setup that we used in the experimental evaluation.

Furthermore, we are currrently conducting two additional experiments that the reviewers suggested and will share the results as soon as we have them, as well as include them in our paper:
**1) Investigating the benefits of context from multiple tables:** As suggested by reviewer HLok, we plan to compare a model trained on single tables compared to one trained on multiple tables to show benefits of having more context. For the evaluation, we will use the table name prediction task.
**2) Investigating the benefits of a model pre-trained on our corpus for a downstream task:** We additionally plan to evaluate a T5 model that has been pre-trained on our corpus for the downstream task and compare it to a model that is trained from scratch for this task. We will also compare to a vanilla T5 model as suggested by reviewer Gs1a. For the evaluation, we will use the task of semantic column type annotation for which task-specific models and data sets exist.

We provide more details in the response to each reviewer. We hope the answers help with the main questions outlined by the reviewers and we are more than happy to answer any further questions.

---

> ### Author Rebuttal · Authors · 2024-08-27
>
> To empirically demonstrate the usefulness of our corpus, we have conducted two additional experiments which we will include in the CR version of the paper.
>
> ----
>
> **(1) Benefits of Pre-Training on WikiDBs for a Downstream Task**
>
> We compare T5 pre-trained on WikiDBs to vanilla T5 on the task of *column type annotation (CTA)*, a task often used in the literature [1-4] where the goal is to annotate each table column with a semantic column type from a pre-defined ontology (e.g., DBpedia). Since there is no multi-table evaluation dataset for CTA, we use a corpus based on individual tables to show the effects of pre-training with WikiDBs.
>
> In particular, for the experiment, we use the GitTablesCTA benchmark [5] also used in the 2021 SemTab challenge [6]. We pre-train T5-3B on WikiDBs as described in Section 5.1. Afterward, we fine-tune the pre-trained T5-3B as well as the vanilla T5-3B model for 50 epochs on GitTablesCTA to predict the column type of a specified column based on the first three rows of the table.
>
> **Results: T5 pre-trained on WikiDBs achieves a validation accuracy of 0.44, whereas vanilla T5 only achieves an accuracy of 0.41. Moreover, Figure 1 in the attached PDF shows that T5, when pre-trained on WikiDBs, requires fewer fine-tuning steps to achieve higher accuracies. For example, after the first 100 training steps, T5 pre-trained on WikiDBs achieves a maximum validation accuracy of 0.37, whereas vanilla T5 only achieves a validation accuracy of 0.25.**
>
> ----
>
> **(2) Benefits of the Multiple-Table Context [Preliminary]**
>
> To investigate the benefits of the multi-table contexts in WikiDBs, we train a single-table and a multi-table model using the GNN-based architecture from [7] with T5 as LLM on the task of *table name prediction*. We initialize both models (single-table and multi-table) with the same pre-trained checkpoint of T5-3B and use a training setup similar to that described in Section 5.1 to train the GNN. For the training, we use a batch size of 32 and a learning rate of 3.2e-4 and train for 100 epochs.
>
> Since pre-training the multi-table model takes over a week, we had to restrict the graph context for this preliminary experiment to only include the table schema (column names of the main table and table names of the connected tables).
>
> **Results: In our very initial experiment, our multi-table model already provides benefits over the single-table model without much tuning w.r.t. how many of the neighboring tables we add as context. To be more precise, the multi-table model achieves an exact match accuracy° of 0.1840 and a token-wise accuracy of 0.2834, whereas the single-table model only achieves an exact match accuracy of 0.1800 and a token-wise accuracy of 0.2607, already indicating the benefit of the multi-table context over the single-table context. We expect this gap to further increase as our understanding of multi-table model architectures and training processes improves.**
>
> ----
>
> We hope that these two initial experiments demonstrate the use of WikiDBs for the pre-training of tabular foundational models as well as how multiple tables can be beneficial as context for such models.
>
> Moreover, we believe that corpora like WikiDBs are an important starting point that will enable further research into how different tabular data understanding tasks are impacted by the presence of more context in the form of multiple relational tables, as well as what the architectures of such models should look like.
>
> We are happy to answer any further questions.
>
> ----
>
> *°Exact match accuracy (EM accuracy) measures how often the predicted table name is exactly the same as the ground truth table name. By contrast, the more relaxed token-wise accuracy considers how many of the individual tokens of the predicted table name match those of the ground truth table name. For example, the predicted table name “BasketballPlayer” for the ground truth “BasketballPlayerStatistics” results in an EM accuracy of 0, but a token-wise accuracy of 0.67 since two of the three tokens are correct.*
>
> References:
>
> [1] M. Hulsebos u. a., „Sherlock: A Deep Learning Approach to Semantic Data Type Detection“, in Proceedings of the 25th ACM SIGKDD International Conference on Knowledge Discovery & Data Mining, in KDD ’19. New York, NY, USA: Association for Computing Machinery, Juli 2019, S. 1500–1508. doi: 10.1145/3292500.3330993.
>
> [2] D. Zhang, M. Hulsebos, Y. Suhara, Ç. Demiralp, J. Li, und W.-C. Tan, „Sato: contextual semantic type detection in tables“, Proc. VLDB Endow., Bd. 13, Nr. 12, S. 1835–1848, Aug. 2020, doi: 10.14778/3407790.3407793.
>
> [3] S. Langenecker, C. Sturm, C. Schalles, und C. Binnig, „Pythagoras: Semantic Type Detection of Numerical Data Using Graph Neural Networks“, Proceedings of the 27th International Conference on Extending Database Technology (EDBT), 25th March-28th March, 2024, 2023.
>
> [4] K. Korini und C. Bizer, „Column Type Annotation using ChatGPT“, 30. Juli 2023, arXiv: arXiv:2306.00745. doi: 10.48550/arXiv.2306.00745.
>
> [5] M. Hulsebos, Ç. Demiralp, und P. Demiralp, „GitTables benchmark - column type detection“. Zenodo, November 2021. doi: 10.5281/zenodo.5706316.
>
> [6] https://www.cs.ox.ac.uk/isg/challenges/sem-tab/2021/index.html
>
> [7] L. Vogel, B. Hilprecht, und C. Binnig, „Towards Foundation Models for Relational Databases [Vision Paper]“, 24. Mai 2023, arXiv: arXiv:2305.15321. doi: 10.48550/arXiv.2305.15321.

---

> > ### Comment · Reviewer_HLok · 2024-08-27
> >
> > Thanks for all the hard work to show the empirical contribution.  I've adjusted my scores accordingly.

---

### Decision · Program_Chairs · 2024-09-26

**Decision:**

Accept (Spotlight)

**Comment:**

The authors introduce a benchmark containing 100K relational databases with multiple tables that are linked together with foreign key relationships.  The preparation of the dataset is detailed and explained well. Experimental results showcase some interesting preliminary results about how the effectiveness of LLMs as they are applied to tabular data, and highlight that further work is needed to replicate the performance that is currently achieved on text.